# Haplotypes of the *tRNAleu-COII* mtDNA Region in Russian *Apis mellifera* Populations

**DOI:** 10.3390/ani13142394

**Published:** 2023-07-24

**Authors:** Milyausha D. Kaskinova, Luisa R. Gaifullina, Elena S. Saltykova

**Affiliations:** Institute of Biochemistry and Genetics, Ufa Federal Research Center, Russian Academy of Sciences, Prospekt Oktyabrya 71, 450054 Ufa, Russia; lurim260578@gmail.com (L.R.G.); saltykova-e@yandex.ru (E.S.S.)

**Keywords:** *tRNAleu-COII* loci, hybridization, M lineage haplotypes, C lineage haplotypes

## Abstract

**Simple Summary:**

One of the most discussed issues in beekeeping is the introduction and hybridization of the honey bee subspecies *Apis mellifera mellifera* (dark forest bee). In most of Russia, the dark forest bee is a native subspecies. However, due to hybridization with introduced subspecies, a small number of purebred populations of the dark forest bee remained. In this study, using the mitochondrial marker *tRNAleu-COII*, we established the maternal descent of honey bees from 19 regions of Russia. As a result, it was found that 198 of the studied colonies belong to the M evolutionary lineage and 71—to the C lineage. We have found two different population groups of the dark forest bee in Russia—one of them belongs to the haplogroup M17, the other to M4’. Haplogroup M4′ dominated in the European *A. m. mellifera* and *A. m. iberiensis* populations. Whereas haplogroup M17 was rare in European populations of the dark forest bee.

**Abstract:**

Analysis of the mtDNA *tRNAleu-COII* locus is a widely used tool to establish belonging to a particular evolutionary lineage of *Apis mellifera* L. (lineages A, M, C, O, and Y). In Russia, most of the area was once inhabited by *Apis mellifera mellifera* from the M evolutionary lineage, but the introduction of bee subspecies from the southern regions of Russia (*A. m. caucasica*, *A. m. carnica*) and from abroad (*A. m. carnica*, *A. m. ligustica*) led to fragmentation of their native range. In this study, the results of assessing the haplotype number for the *tRNAleu-COII* locus of mtDNA in Russian *Apis mellifera* populations were presented. We analyzed 269 colonies from 19 regions of Russia. As a result, two evolutionary lineages were identified: the East European lineage C (26.4%) and the Northwestern European lineage M (73.6%). A total of 29 haplotypes were identified, 8 of them were already reported, and 21 were found to be novel. From the C lineage, haplotypes C1, C2, C2c, C2j, and C3 were predominant. All M lineage samples from Russia belong to the M17 and M4’ haplogroups but have only minor variations in the form of nucleotide substitutions. An analysis of publications devoted to the *tRNAleu-COII* locus haplotypes, as well as an analysis of the available *tRNAleu-COII* sequences in GenBank, showed that there is still a problem with the haplotype nomenclature.

## 1. Introduction

One of the most discussed issues in beekeeping is the introduction and hybridization of honey bee subspecies [1,2]. If the natural hybridization of subspecies is the driving force behind speciation and is necessary to maintain the genetic diversity of populations [3], then hybridization caused by human intervention can lead to the loss of the native gene pool and even the extinction of the species [4,5,6]. Hybridization of honey bee subspecies can lead to the extinction of subspecies or local populations. As, for example, the infamous case of the complete replacement of the native population of the dark forest bee *Apis mellifera mellifera* by the Italian bee *A. m. ligustica* in Germany [1]. Among the European subspecies, the dark forest bee L. suffered the most from hybridization, and introgression of the gene pool of other subspecies is the greatest danger for this subspecies [1,2].

In most of Russia, the dark forest bee [7,8] is a native subspecies. It is the main source of breeding material. However, due to hybridization with introduced subspecies, a small number of purebred populations of the dark forest bee remained. In recent years, several surviving populations of *A. m. mellifera* have been discovered in the Republic of Bashkortostan, the Perm Krai, the Republic of Udmurtia, etc. [9,10]. In the Southern and North Caucasian federal districts, the native subspecies is the gray mountain Caucasian bee, *A. m. caucasica*. Subspecies *A. m. carnica* and *A. m. ligustica*, as well as Buckfast hybrids, are imported into Russia from European and Central Asian countries [7,9,10].

The problem of preserving the dark forest bee, and indeed the purebred gene pool of different subspecies of the honey bee, was raised due to the fact that hybridization leads to the loss of gene associations characteristic of each particular subspecies and ecotype, responsible for adaptability and ecological plasticity. This prompted scientists from different countries to search for a reliable method for identifying the subspecies of honey bees and taking measures to prevent hybridization [2,9,11,12,13,14].

There are more than 30 subspecies of *A. mellifera* [8,15,16,17]. Based on morphometric differences, these subspecies were initially divided into four evolutionary lineages: African (A), West and North European (M), East European (C), and West and Central Asian (O) [8]. *A. m. mellifera* belongs to the evolutionary lineage M, along with the Iberian bee *A. m. iberiensis* and the Xinyuan bee *A. m. sinisxinyuan*. Based on this classification, *A. m. caucasica* belongs to the evolutionary lineage O, with the subspecies *A. m. carnica* and *A. m. ligustica*—to the C lineage. Genome-wide data [17,18,19] also confirm these findings. To study the genetic diversity of honey bees and to differentiate evolutionary lineages, the DraI test was also developed [20]. This test is based on the analysis of the lengths of restriction fragments of the *tRNAleu-COII* (or *COI-COII*) intergenic locus. It can be used to differentiate more than 100 haplotypes from five evolutionary lineages [21,22,23,24,25]. The subspecies *A. m. yemenica* was assigned to the fifth evolutionary lineage Y [22]. *A. m. caucasica*, which, based on morphometric and genome-wide data, belongs to the lineage O, has common *tRNAleu-COII* haplotypes with subspecies from the evolutionary lineage C (*A. m. ligustica, A. m. carnica*) [26]. That is, this test cannot be used to differentiate lineages C and O.

Despite its limitations, analysis of the mtDNA *tRNAleu-COII* intergenic locus remains one of the available methods for establishing evolutionary lineages and the maternal origin of honey bee populations. The high polymorphism of the *tRNAleu-COII* locus makes it a convenient marker for solving applied problems in beekeeping [9,27,28]. This intergenic locus is a non-coding sequence consisting of a part of the *tRNAleu* gene, elements conventionally designated as P and Q, and a part of the *COII* gene. The P element has four forms: P0, P, P1, and P2, which differ from each other by insertions and deletions. The Q element is a tandem repeat. Subspecies from the evolutionary lineage M are characterized by the P(Q)_1–n_ variant. In lineage A, element P differs from that in lineage M by the presence of a 13 bp insert (AAACAAAATATAA) and is denoted as P0 [20]. Moreover, in some subspecies from the A lineage, the P1 element is found, which differs from P by a 15 bp deletion. Representatives of the evolutionary lineage C lack the P element, and the locus is represented as part of the *tRNAleu* gene, one Q element, and part of the COII gene. Subspecies from the Y lineage are characterized by the P2 form (18 bp deletion in the P element) [22,29]. A characteristic feature of the *tRNAleu-COII* locus is the presence of DraI (TTTAAA) restriction sites. Using the DraI restriction enzyme, it is possible to distinguish haplogroups of the *tRNAleu-COII* locus, such as C1, C2, C3, M4, M7, A1, etc. [27,30,31]. Within the haplogroup, haplotypes are distinguished that differ from each other by small substitutions but have the same DraI restriction sites. Haplotypes can be determined using sequencing. For example, within the haplogroup C2, haplotypes C2, C2c, C2j, C2d, etc. are distinguished. Within the haplogroup M4, haplotypes M4s, M4e, M4d, etc., are distinguished [27,31].

In this study, we aimed to investigate *tRNAleu-COII* locus haplotypes in Russian honey bee populations. It has already been said that the territory of Russia is inhabited by subspecies from three evolutionary lineages: M (*A. m. mellifera*), C (*A. m. carnica, A. m. ligustica*), and O (*A. m. caucasica*). Although we cannot distinguish the C and O evolutionary lineages using *tRNAleu-COII* locus analysis, we would be interested to know if there are differences in haplotype frequencies between *A. m. caucasica* and subspecies from the lineage C. Because earlier, Alburaki et al. [31] showed that *A. m. caucasica* is characterized by the C2j haplotype. The Russian population of *A. m. mellifera* has not yet been assessed for the *tRNAleu-COII* haplotypes. Previously, data for Russian populations were obtained only on the basis of differences in the lengths of *tRNAleu-COII* variants detected in PAAG without sequencing [9,10,32]. In addition, it is known that in European *A. m. mellifera* populations, the predominant haplotypes are M4 and M4’ [27]. We will try to find out how close European and Russian populations of the dark forest bee are.

## 2. Materials and Methods

### 2.1. Sampling and DNA Extraction

Worker bees were selected inside hives from 269 colonies from 19 regions of Russia between 2020 and 2021. On the map (Figure 1), the numbers indicate the regions where samples were taken.

DNA was isolated from the thorax muscles using the DNA-Extran-2 kit (Syntol, Moscow, Russia). The quality and quantity of isolated DNA were measured on an Implen N50 spectrophotometer. 

### 2.2. tRNAleu-COII Locus Analysis

PCR analysis of the *tRNAleu-COII* intergenic locus was performed using primers E2 (5′-GGCAGAATAAGTGACATTG-3′) and H2 (5′-CAATATCATTGATGAACC-3′) [20] in a final volume of 20 μL: 15 μL sterile ddH2O, 2 μL of 10× PCR Buffer, 0.4 μL dNTP, 0.6 μL each primer (10 pmol/μL), 0.3 μL Taq DNA polymerase, and 2 μL DNA template. PCR conditions: initial denaturation at 94 °C for 5 min, followed by 30 cycles of denaturation at 94 °C for 30 s, annealing at 50 °C for 30 s, and elongation at 72 °C for 1 min, with a final elongation at 72 °C for 10 min. PCR products were examined on 8% polyacrylamide gels (PAAG) stained with ethidium bromide. The gels were visualized in a Gel Doc™ XR+ photosystem (BioRad, Hercules, CA, USA). As a result of amplification of the *tRNAleu-COII* locus, PCR products of about 600, 800, and 1000 bp in size were obtained, corresponding to variants Q, PQQ, and PQQQ (Figure 2).

Sequencing was carried out using the Applied Biosystem sequencer at the Syntol Company (Moscow, Russia). Each sample was sequenced twice. Nucleotide sequences were manually edited in Mega 5.2 [33] software to produce the consensus sequences, which were then aligned with previously published sequences using the ClustalW algorithm. The phylogenetic tree was constructed using the maximum likelihood method based on the Kimura 2-parameter model in Mega 5.2.

*tRNAleu-COII* locus sequences are available at the link https://doi.org/10.6084/m9.figshare.22348063 (accessed on 17 July 2023). All unique sequences are given in the Appendix A. The haplotype network was predicted using a median-joining algorithm using the PopArt v1.7 software package (https://popart.maths.otago.ac.nz/ accessed on 20 January 2023). In silico DraI test of the *tRNAleu-COII* locus and restriction fragment length calculation (Appendix A) were performed in Unipro UGENE ver. 36.

## 3. Results

### 3.1. tRNAleu-COII Haplotypes from Evolutionary Lineage C

Out of 269 colonies, 71 belong to the evolutionary lineage C (allelic variant Q, 571 bp). Among them, seven already known C haplotypes were identified: C1, C2, C2c, C2j, C2ja, C2l, C3, and six new haplotypes, which were named C2i2, C2jf, C2jd, C2je, C1j, and C4a (Appendix A). Haplotype C1, often found in *A. m. ligustica*, was identified in six colonies from the Republic of Adygea and in one colony from the Novgorod Oblast. Haplotype C1j from Adygea, according to the in silico DraI test (Table 1), also belongs to haplogroup C1. Colonies from Adygea were declared by beekeepers as *A. m. ligustica* (queens were purchased in Italy and the USA) and three of them as Cordovan mutants. Haplotype C2 was identified in nine colonies from the Ryazan Oblast, the Primorsky Krai, the Republic of Adygea, the Sverdlovsk Oblast, and the Tver Oblast. The haplotype C2c had the highest frequency—it was found in 28 colonies. According to the generalized data of Tanaskovic et al. [34], this haplotype, along with the C2d, C1a, and C2e haplotypes, is the most common in colonies from the C lineage. Haplotype C2j was identified in 12 colonies from the Krasnodar Krai (10 colonies), the Republic of Adygea (one colony), and the Omsk Oblast (one colony). Haplotype C3 was identified in three colonies from the Ryazan Oblast and Bashkortostan. Haplotype C2l was identified only in one colony in the Novgorod Oblast (number 6 in Figure 1). New haplotypes also had a low frequency. In silico DraI test showed another haplogroup called C4, which includes one haplotype called C4a (also from the Novgorod Oblast).

In silico DraI test (Table 1) showed that C2i2, C2jf, C2jd, and C2je belonged to the C2 haplogroup. Seven of our selected samples of *A. m. caucasica* from the Krasnopolyansk Experimental Station of the Krasnodar Krai belong to the C2j haplotype, and three samples belong to the C2jf, C2jd, and C2je haplotypes. Haplotypes C2jd and C2je differed from C2j by one substitution. The BLAST search showed that haplotype C2jf is closest to haplotype C2j. GenBank *tRNAleu-COII* sequences belonging to *A. m. caucasica* (Ap018404.1 from Russia, OP404074.1 and OP404073.1 from the Republic of Turkey, and MN714160.1 from the USA) belong to the evolutionary lineage C. These sequences belong to the C2j haplotype, only MN714160.1 differs from the C2j haplotype by 1 substitution. 

The BLAST search for the C2i2 haplotype sequence showed that the C2i haplotype is closer to it. C2i2 differs from C2i by two nucleotide substitutions. The C2i1 (MH939344) haplotype differs from C2i by one substitution.

On the basis of morphometric parameters and nuclear DNA markers, *A. m. caucasica* belongs to the evolutionary lineage O. The lineage O also includes the subspecies *A. m. anatoliaca*, *A. m. remipes*, *A. m. macedonica*, *A. m. cecropia*, and *A. m. cypria*. *tRNAleu-COII* sequences of *A. m. anatoliaca* and *A. m. macedonica* available in GenBank also belong to the C lineage haplogroups. For example, the sequences of *A. m. anatoliaca* ON933877.1, MN701760.1-MN701763.1 belong to haplotype C2, and FJ357798.1—to C1. The haplotypes of *A. m. macedonica* differ by one substitution from the C2 haplotype. For *A. m. remipes*, *A. m. cecropia*, and *A. m. cypria*, we did not find *tRNAleu-COII* sequences. Therefore, our data also confirmed that the *tRNAleu-COII* test does not differentiate between C and O evolutionary lineages.

Figure 3 shows the median-joining network of haplotypes from the evolutionary lineage of C.

### 3.2. tRNAleu-COII Haplotypes from Evolutionary Lineage M

Subspecies from the evolutionary lineage M have allelic variants P(Q)_1-n_. Out of 269 colonies, 31 had the PQQQ allelic variant (1020 bp) and 167 colonies had the PQQ variant (825 bp). A total of 12 M (PQQ) and four M’ (PQQQ) haplotypes were identified. Comparison with the M haplotypes from GenBank showed that only one haplotype belongs to the M4s haplotype (MW939595.1), previously identified in Poland [25]. The remaining haplotypes are new and have not been found in European populations of *A. m. mellifera*. The number of nucleotide differences between M haplotypes ranged from 1 to 8 (*A. m. mellifera* assembly GCA_003314205.2 as reference).

We performed an analysis of publications devoted to the *tRNAleu-COII* locus haplotypes, as well as an analysis of the available *tRNAleu-COII* sequences in GenBank, and found that the problem with haplotype nomenclature still exists. There have been several attempts to revise the haplotypes and formulate rules for their nomenclature that would avoid incorrect names for new haplotypes [27,31,35]. For example, in our case, one of the haplotypes belongs to the previously identified M4s (MW939595.1) haplotype, but the size of the restriction fragments for this sequence corresponds to the M17 haplogroup [35]. 

Our M haplotypes were cut at four restriction sites into fragments of length 142/66/131/65/422 (PQQ). The same restriction profile was found in M4s, M4sa, M4sc, M4na, M4pa, M4b, M4r, M4rb, M4rc, M4t, and M4ta [25]. Since the classification of *tRNAleu-COII* haplotypes is based on the number of DraI restriction fragments, all M lineage samples from Russia belong to the M17 haplogroup [35]. Therefore, based on the rules of nomenclature, we gave the following names for our M17 haplotypes: M17j, M17n, M17k, M17o, M17p, M17q, M17l, M17r, M17s, M17m, M17t, and M17u.

Errors in the nomenclature were observed for the haplotype M4. M4 haplotypes in GenBank are represented under accession numbers HQ337436.1, FJ743637.1, KX463886.1, and FJ478006.1. The last three haplotypes differ from the first one by the presence of a 60 bp insertion and single nucleotide substitutions. In addition, fragment sizes at DraI restriction sites, characteristic of M4 (142/65/131/65/422), were found only in KX463886.1 and FJ478006.1. These two sequences differ by one C/T nucleotide substitution at position 652. The FJ743637.1 sequence belongs to the M7 haplotype in terms of the size of the restriction fragments (47/95/65/131/65/422). The HQ337436.1 sequence was cut at four restriction sites into fragments of length 100/65/130/65/29/289, and, moreover, in a 130 bp fragment, there was an “N” base.

The M17j haplotype had the highest frequency; it was found in 146 colonies (Figure 4). The M17k haplotype was identified in nine colonies from the Republic of Udmurtia, the Republic of Bashkortostan, and the Perm Krai. The remaining haplotypes occurred singly or twice. Five new haplotypes (M17n, M17o, M17r, and M17u in Figure 4) that met once were identified in the Burzyansky district of Bashkortostan. This district is a well-known dark forest bee reserve, where log hive beekeeping has been preserved [7,9,32]. 

Phylogenetic analysis of our sequences and sequences from GenBank using the maximum likelihood method (Appendix A) showed that M haplotypes from Russia form one cluster with *A. m. mellifera* from Poland [25]. The restriction profile of these sequences also corresponds to the M17 haplogroup (142/66/131/65/422). At the same time, we see that the studied M17 haplotypes and the haplotypes from Poland do not form a common cluster on the tree with the M17 haplotype from GenBank (HQ337450), despite the common restriction profile (Appendix A).

Four haplotypes M’ (PQQQ) were found in colonies from the Republic of Bashkortostan (Yanaulsky district), Pskov, Novgorod, Sverdlovsk, and Leningrad Oblast. In silico DraI test showed that our M’ haplotypes have six restriction sites and are cut into fragments of length 142/65/131/65/131/65/421 (corresponding to the PQQQ allele). Therefore, these sequences belong to the M4’ haplogroup (haplotypes M4h’, M4g’, M4i’, and M4j’).

The M4g’ haplotype had the highest frequency—it was identified in 23 colonies from four regions of Russia. Therefore, in accordance with the rules of nomenclature [35], we named this haplotype M4g’. M4h’ was identified in six colonies from two regions. M4i’ and M4j’ met singly in the Yanaul region of Bashkortostan. The number of nucleotide differences between M4’ haplotypes ranged from 1 to 5 (as reference M4g’). Between M4g’ and M4h’, four nucleotide substitutions and a deletion of one nucleotide were detected. M4i’ and M4j’ differed from M4g’ by two and one substitution, respectively. Phylogenetic analysis of M’ haplotypes using the maximum likelihood method (Appendix A) also showed that *A. m. mellifera* from Russia and Poland and *A. m. iberiensis* from Spain form a common cluster [25,36]. Moreover, they did not form a common cluster with M4’ haplotypes from GenBank. 

Allelic variants of PQQQ (1020 bp), PQQ (825 bp), and Q (571 bp) are clearly distinguishable in an 8% polyacrylamide gel (Figure 2), which allows, without sequencing or DraI test, to establish belonging to the evolutionary lineage M. This rule works only for populations where there is no introduction of subspecies from the evolutionary lineages A and Y, which can have the same fragment size of the *tRNAleu-COII* locus.

## 4. Discussion

The maternal origin of the Russian honey bee (*Apis mellifera* L.) populations was examined through a molecular approach using the mitochondrial DNA *tRNAleu-COII* intergenic locus. Samples from 269 colonies were selected from 19 regions of Russia. As a result, two evolutionary lineages were identified: The East European lineage C (26.4%) and the Northwestern European lineage M (73.6%). A total of twenty-nine haplotypes were identified, eight of them were already reported, and twenty-one were found to be novel. 

We assessed which haplotypes from evolutionary lineage C are common in Russia. Our study revealed the presence of seven haplotypes described previously as C1, C2, C2c, C2j, C2l, C3, C2ja, and six new C haplotypes (C2i2, C2jf, C2jd, C2je, C1j, and C4a). According to the results obtained, the main exported subspecies are *A. m. carnica* and *A. m. ligustica*. In the southern regions of Russia, the native subspecies is the gray mountain Caucasian bee, *A. m. caucasica*. We found that this population is dominated by the C2j haplotype. Alburaki et al. [31] also showed that *A. m. caucasica* is characterized by the C2j haplotype. The Krasnopolyanskaya Experimental Station of Beekeeping, where we took samples of *A. m. caucasica*, was established in 1963 with the aim of preserving and breeding *A. m. caucasica* [37]. To this day, this Station carries out her scientific and economic activities for breeding this subspecies (https://kosp-plem.ru/ assessed on 15 March 23). This subspecies is morphometrically different from C lineage subspecies (*A. m. carnica*, *A. m. ligustica*) [8], which makes it easier to breed. 

The native subspecies in the greater territory of Russia is the dark forest bee, *A. m. mellifera*. It is of particular importance for Russian beekeeping, as it is ideally suited to a short foraging season and a long winter. Therefore, the identification of this subspecies is extremely relevant. Rortais et al. [27] reported 91 M haplotypes, among which the M4 and M4’ haplotypes are the most common. Haplotypes from the M evolutionary lineage found in Russian populations of *A. m. mellifera* belong to haplogroups M17 and M4′.

Since 1996, more than 9000 colonies have been analyzed by our laboratory using the analysis of polymorphisms in the *tRNAleu-COII* locus (without sequencing, only based on the DraI test). According to these results, local populations of *A. m. mellifera* have been identified in the Republic of Bashkortostan, Tatarstan, Udmurtia, Perm Krai, etc. [9,10].

As noted above, subspecies from the evolutionary lineages C/O inhabit the southern regions of Russia—the Southern and North Caucasian federal districts. The uncontrolled transfer of bee colonies from one region of Russia to another and the export of bee packages and queens from other countries have led to the fact that the range of the dark forest bee in the central and northern parts of Russia has become fragmented. The modern range of the dark forest bee in Europe is also fragmented and is represented by local populations in Switzerland [38,39], Denmark [13], Norway [14,40], France [39,41], England [42], Poland [2,25], and Ireland [28].

In Russia, the protection and restoration of the population of the dark forest bee are carried out in the Shulgan-Tash State Nature Reserve (since 1958) and the Altyn Solok Reserve (since 1997). In addition to them, individual beekeepers whose apiaries are located nearby, on their own initiative, unite in cooperatives and breed exclusively *A. m. mellifera* with the scientific support of our laboratory [9,32]. 

In Europe, various organizations and associations have also been created to protect this subspecies. For example, in Great Britain and Ireland in 1964, an association, now known as the BIBBA (Bee Improvers and Bee Breeders Association), was created, whose purpose is the conservation and restoration of the dark forest bee [42]. Based on a polymorphism analysis of 11 microsatellite loci, Jensen et al. (2005) [1] found from 4 to 10% introgression of *A. m. ligustica* alleles in the populations of the dark forest bee in the countries of Scandinavia and the British Isles. In Sweden, in 1990, the Nordbi Association was organized (https://www.nordbi.se/ assessed on 15 March 2023). To preserve the dark forest bee in Sweden, artificial insemination and subspecies control based on morphometrics (cubital index and discoidal displacement) and mtDNA analysis were used [14]. In Denmark, the dark forest bee is located on the protected area of the island of Laeso [12,13]. A study [11] confirmed the purity of Laeso island bees. In Switzerland, the association “Swiss Mellifera Bee Buddies” was organized in 1993, and a law was passed in 2008 to protect and breed the native dark forest bee [12,38]. Under their protection were six isolated populations, each of which was confirmed to belong to the dark forest bee. However, high levels of other subspecies introgression have been found in some regions of the country [12,43]. Soland-Reckeweg et al. (2009) [38], based on the analysis of polymorphism at 12 microsatellite loci, revealed a 70% level of introgression of *A. m. carnica* alleles in populations of the Swiss dark forest bee. 

In Finland, the dark forest bee populations are protected both by natural factors (island populations) and anthropogenic factors. According to 1990 data, about 95% of the honey bee population (out of 40,000 colonies) in Finland was the *A. m. ligustica* subspecies. About 10,000 queen bees have been imported to Finland in the last three decades to restore dark forest bee populations [44].

On the island of Palma in the Canary Archipelago, the protection of the dark forest bee has been carried out since 1996. The subspecies of bees was determined using genetic methods [12]. Since 2001, regional laws have been established to control the conservation and breeding of the dark forest bee [12].

Morphometric analysis of Polish bees showed the presence of preserved populations of *A. m. mellifera* [45]. The existence of at least two of them in Poland has been confirmed by DNA analysis. Oleksa et al. [2], using polymorphism data from nine microsatellite loci, showed up to 30% introgression of *A. m. carnica* alleles in dark forest bee populations.

In Norway, dark forest bee populations also survive in protected areas. In the 20th century, subspecies such as *A. m. ligustica*, *A. m. caucasica,* and *A. m. carnica*, as well as Backfast hybrids, were imported to Norway. Beekeepers usually use morphometric methods to differentiate these subspecies. To confirm the results of the morphometric analysis, genetic methods were used. Analysis of the Norwegian honey bee population using 25 microsatellite loci confirmed that it belonged to *A. m. mellifera* [40]. The Association of Norwegian Beekeepers is responsible for the national breeding programs of *A. m. carnica* and *A. m. mellifera* in isolated areas. In an area of about 3500 km^2^, only *A. m. mellifera* was allowed to be kept. Approximately the same territories were allocated for the breeding of *A. m. carnica*.

*A. m. mellifera* protected areas have been established in several regions of France [27]. Analysis of microsatellite loci showed that French populations of dark forest bees are usually introgressed by the C lineage gene pool [32].

The common marker used to identify the dark forest bee in all of the above countries was the *tRNAleu-COII* intergenic locus [27]. The analysis of the *tRNAleu-COII* locus is the starting point for further analysis of honey bee populations. The simplicity and low cost of this method make it a convenient tool for searching for and restoring *A. m. mellifera* populations. However, despite the fact that a nomenclature for the *tRNAleu-COII* haplotypes has been developed [35], there is still a problem with their erroneous naming. The effort of all engaged researchers in the repeated and final revision of the *tRNAleu-COII* haplotypes is required.

## 5. Conclusions

In this study, we reported for the first time sequences of *tRNAleu-COII* haplotypes from the Russian honey bee populations. We have identified one previously described haplotype, eleven new M17 haplotypes, and four M4’ haplotypes. Thus, we found two different population haplogroups of *A. m. mellifera* in Russia—M17 (PQQ, 142/66/131/65/422) and M4’ (PQQQ, 142/65/131/65/131/65/421). Moreover, we showed that *A. m. caucasica* from Krasnopolyanskaya Experimental Station of Beekeeping is characterized by the C2j haplotype.

Very interesting is the large number of haplotypes in haplogroup C2. At least 40 different C2 haplotypes are known, while the C1 and C3 haplogroups cannot boast of such diversity. We have identified another haplogroup called C4, which includes one haplotype called C4a. The high diversity of the C2 haplogroup and the presence of misclassified haplotypes pose a new research challenge.

## Figures and Tables

**Figure 1 animals-13-02394-f001:**
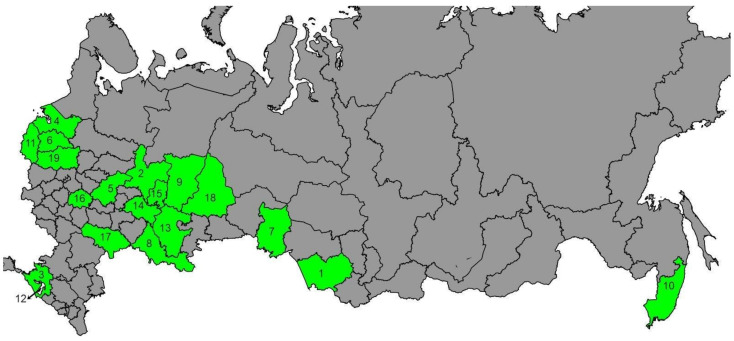
Map of the sampling sites. The regions where the bees were selected are marked in green. The numbers indicate the regions where samples were taken: 1—Altai Krai, 2—Kirov Oblast, 3—Krasnodar Krai, 4—Leningrad Oblast, 5—Nizhny Novgorod Oblast, 6—Novgorod Oblast, 7—Omsk Oblast, 8—Orenburg Oblast, 9—Perm Krai, 10—Primorsky Krai, 11—Pskov Oblast, 12—Republic of Adygea, 13—Republic of Bashkortostan, 14—Republic of Tatarstan, 15—Republic of Udmurtia, 16—Ryazan Oblast, 17—Saratov Oblast, 18—Sverdlovsk Oblast, 19—Tver Oblast.

**Figure 2 animals-13-02394-f002:**
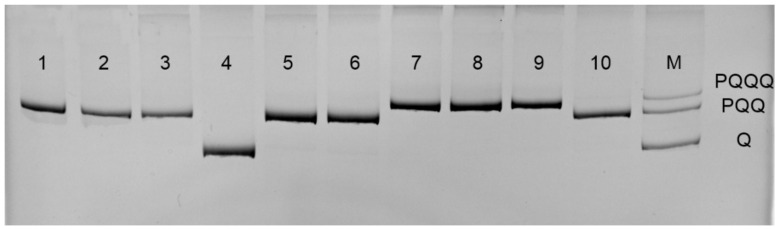
PCR products of the mtDNA *tRNAleu-COII* locus, where 1–10—samples, M—marker.

**Figure 3 animals-13-02394-f003:**
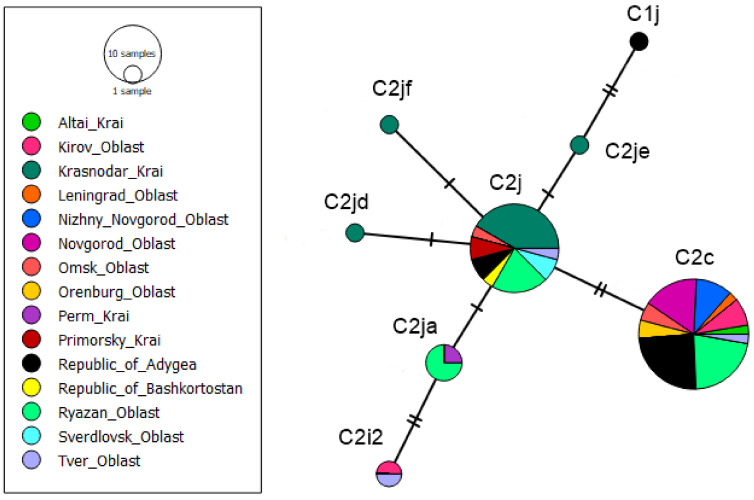
Median-joining networks among the C haplotypes of *Apis mellifera*.

**Figure 4 animals-13-02394-f004:**
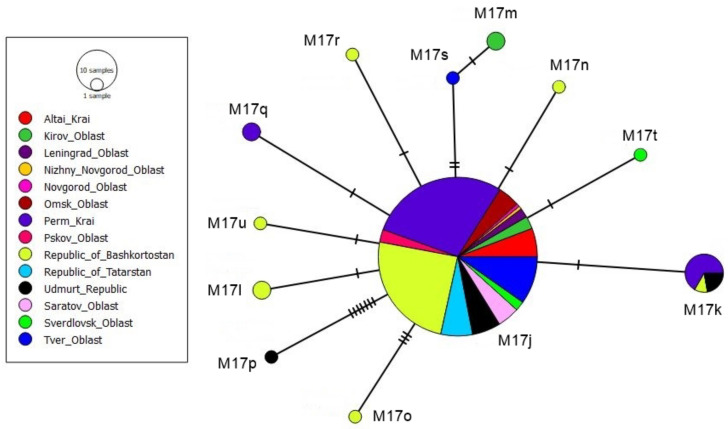
Median-joining networks among the M haplotypes of *Apis mellifera*.

**Table 1 animals-13-02394-t001:** DraI restriction sites for the *tRNAleu-COII* locus in the study sample.

DraI Restriction Sites	Haplotypes	Haplogroup
47/40/64/420	C2, C2c, C2l, C2j, C2ja, C2i2, C2jf, C2jd, C2je	C2
47/41/64/420	C1, C1j	C1
47/40/63/420	C3	C3
47/39/64/420	C4a	C4

## Data Availability

Data has been deposited in the Figshare repository. *tRNAleu-COII* locus sequences are available at the link https://doi.org/10.6084/m9.figshare.22348063 (accessed on 17 July 2023).

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
