# Peer review of "Haplotypes of the tRNAleu-COII mtDNA Region in Russian Apis mellifera Populations"

_animals, 2023, doi:10.3390/ani13142394_

Round 1

Reviewer 1 Report

The paper entitled “haplotype diversity of the tRNAleu-COII intergenic mitochondrial DNA locus in Apis mellifera populations in Russia” uses a popular region of the mtDNA to study the genetic diversity patterns in Russia. It is interesting mainly because of the geographic region, but there are several problems that need to be solved before publication. The manuscript contains some analyses that do not add meaningful information, such as the NJ networks, and on the other hand, other analyses that could be more informative are missing. For instance, the paper is called haplotype diversity. However, there are no diversity calculations in the manuscript and this is an important analysis for this kind of paper. The English needs to be revised and the information better organized. Most of the information that is in the discussion should be in the introduction. The paper also does not refer to the drawbacks of this intergenic region. The methods are also not very clear.

References needed: L40, L44, L45, L49, L51, L123, L219

Title: The title can be improved, I think that the authors do not need to write “tRNAleu-COII intergenic mitochondrial DNA locus” for instance just tRNAleu-COII mtDNA region or just mtDNA intergenic region.

Simple Summary:

Line 9: The authors should say the scientific name of the subspecies

Line 11: The authors should refer to the region.

Line 11: “we established the origin” it is not 100% accurate, the authors study the genetic diversity patterns.

Lines 12-13: The authors also found C lineage. So, this sentence is also not 100% accurate

Abstract

Line 15-16: This sentence is a bit strange and also not accurate. This region studying the genetic diversity

Line 18-19: The information would be more complete if the authors say the lineage of all subspecies

Introduction

The authors should explain better what DraI test is and the relation with tRNAleu-COII intergenic. They have this information but not organized. Also, they should explain how is possible to differentiate the different haplotypes

Line 32: In the case of honey bees is more correct to say that hybridization can lead to the extinction of subspecies or local populations. The sentence “Hybridization can lead to the absorption of one subspecies by another” is not needed.

Line 56: and about the subspecies Apis mellifera sinisxinyuan? The authors should correct the sentence

Line 58: This sentence should be improved. The DraI test was developed to study genetic diversity, of course, also implicitly to differentiate the evolutionary lineages.

Lines 58-63: The authors have to improve this information. For someone that never used the DRAI test this is very difficult to follow.

Lines 66-68: There are several drawbacks of using this marker, one of them is that only tells the maternal history. The authors should explore and write in a clearer way the advantages and disadvantages of this marker.

Line 68. This marker is actually not the best for phylogenetic studies see https://doi.org/10.1007/s13592-019-00632-9 .

Line 70: If the aim of the paper is “investigate the haplotype diversity” some diversity measures should be calculated.

Lines 74-75: I am not sure if I understand this objective. If this region is not able to differentiate these lineages…

Line 76: And about the studies number 9, https://doi.org/10.1134/S1022795422010045

Material and Methods

If the authors want to use this region they should use the nomenclature system that they can see here https://doi.org/10.1007/BF02125651 and that was revised here https://doi.org/10.1007/s13592-017-0498-2 and here https://doi.org/10.1007/s12686-010-9351-x.  Does not make sense the names that they give to the sequences.

The authors also need to revise what is a variant and, haplotype and haplogroup

All the new sequences should be submitted to Genbank.

I understand that the authors performed the NJ analysis and the phylogenetic analysis to answer the objective “to find out how close European and Russian populations of the dark forest bees are”. However, they also want to describe the diversity, and no analysis was performed. See how the following paper calculates the diversity using this region:

https://doi.org/10.1007/s13592-017-0498-2

This region is particularly difficult to use to do phylogenetic and phylogeographic analysis. It is not clear how the authors handled the gaps that are characteristic of this region.

Lines 84-89 This information should be in the legend of Figure 1

Line 100: reference with a different format

Results

Again, the authors should use the nomenclature system for this region.

The authors should describe the median-joining results

It is not clear why the authors did 2 NJ one for C and another one for M, to be honest, I think that this does not add information to the manuscript

The authors should provide a better overview of the haplotypes found in the different regions of Russia that were studied.

Line 132: in one colony, where?

Line 146: there is no description of the sequences that were used by the authors for the comparison in the material and methods section

Line 202-244: The authors should be careful with this sentence.  DraI test is based on PCR amplification of the intergenic region between tRNAleu and COX2 followed by digestion with DraI restriction enzyme. The variants are identified considering the length and restriction site polymorphisms of this region

Discussion

Most of the information that is the discussion should be in the introduction

Line 233: haplotype or lineage?

Line 251: Species in italic

Line 283: The numbers up to 10 should be spelled out. The authors should check this in the manuscript

Figure S1: This tree can be improved, instead of having these big names that make it difficult to see the tree. Why not use only the accession number

Moderate editing of English language to Extensive editing of English language required

Author Response

Dear Reviewer,

Thank you for your careful reading of our manuscript. Thanks to your comments, we hope the article has improved. The table contains all the answers to your comments.

Reviewer 2 Report

I found this article interesting and it could be useful in the context of analyse genetic diversity of honey bee population in Europe.

I suggest some minor revision and a minor editing of English language as well.

Line 64: please rephrase this sentence

Line 251: should be Apis m. mellifera

Line 310: please rephrase the paragraph

I suggest some minor revision and a minor editing of English language as well.

Author Response

Dear Reviewer,

Thank you for your comments, the work is corrected according to your recommendations

  1. corrected according to your comments
  2. corrected according to your comments
  3. corrected according to your comments

Reviewer 3 Report

This study were to focus on the native bees the dark forest in Russia. The study established the origin of honey bees from 19 regions of Russia, and found two different population groups of the dark forest bee in Russia - one of them belongs to the haplogroup M4, the other to M4'. These haplogroups also dominated the European populations of this honey bee subspecies. This research is importatnt to investigate the diversity of honeybees and further protection of honeybees in Russia. I suggest minor revisions of the text.

1. In Figure 1, it is suggested to add the region information of the sampling sites to enhance the readability of the figure.

 2. Fig.4, it is better to edited the figure and cut the edges and the figure would be more beautiful.

Author Response

Dear Reviewer,

Thank you for your comments, the work is corrected according to your recommendations

  1. corrected according to your comments
  2. corrected according to your comments

Round 2

Reviewer 1 Report

The manuscript has shown some improvement; however, there are still some important changes that the authors need to address. I have identified three key areas that require attention:

1) The NJ trees do not add important information, mainly because the authors have not provided a description of the tree results. It is crucial to include a detailed analysis and interpretation of the tree findings.

2) The authors have neglected to perform diversity analysis in the paper, despite its significance and relevance to this type of research. Instead, they have opted to modify the title and objectives of the paper. It is essential to include diversity analysis, as it is not a complex task and provides crucial insights for this study.

3) If the authors intend to use this specific region, they should adhere to the nomenclature system outlined in the following sources: [https://doi.org/10.1007/BF02125651 https://doi.org/10.1007/s13592-017-0498-2]. The cited nomenclature system has been recognized and revised, making it more appropriate for naming the sequences. The current names given to the sequences do not align with this established system and should be corrected.

By addressing these concerns and making the necessary revisions, the authors can further enhance the manuscript and strengthen its scientific rigor.

No comments

Author Response

Dear Reviewer,

Thank you for your perseverance and attention to detail. Thanks to you, we managed to identify an error in the naming of haplotypes. We corrected the manuscript as far as time allowed us. I hope that all comments were taken into account. The answers to your comments are given in the table.

Round 3

Reviewer 1 Report

While I acknowledge the improvements made in the manuscript, I would like to express my opinion regarding the nomenclature used to designate the new haplotypes. In my view, it would be more appropriate to adhere to established nomenclature conventions within the honey bee community. The current nomenclature of HapCR6, HapCR3, HapCR4, and so on does not align with the standard naming system. I suggest that the new haplotypes be designated as C1, C2, C3, and so forth, following the established nomenclature guidelines. 

no comments

Author Response

Dear Reviewer,
The names of the haplotypes were changed in accordance with the rules of nomenclature.
Thank you for your comment.